# Flat Minima in Linear Estimation and an Extended Gauss Markov Theorem

**Simon Segert**
Princeton Neuroscience Institute
Princeton University
`ssegert@princeton.edu`

## Abstract

We consider the problem of linear estimation, and establish an extension of the Gauss-Markov theorem, in which the bias operator is allowed to be non-zero but bounded with respect to a matrix norm of Schatten type. We derive simple and explicit formulas for the optimal estimator in the cases of Nuclear and Spectral norms (with the Frobenius case recovering ridge regression). Additionally, we analytically derive the generalization error in multiple random matrix ensembles, and compare with Ridge regression. Finally, we conduct an extensive simulation study, in which we show that the cross-validated Nuclear and Spectral regressors can outperform Ridge in several circumstances.

## 1 Introduction

Linear models are among the most used of all machine-learning models in applications to science and engineering. In addition to this practical interest, they are also of great theoretical interest. Indeed, the empirical successes of neural networks have proven somewhat at odds with traditional received wisdom from classical statistical learning theory(Belkin et al., 2019). This has begun to be reconciled by careful analysis of well-chosen linear "model systems" such as high-dimensional regression (Hastie et al., 2019), kernel ridge regression (Canatar et al., 2021; Jacot et al., 2020), or linear neural networks (Saxe et al., 2013), which can reproduce many qualitative properties of learning dynamics of neural networks. Thus, the careful study of linear models, especially in the limit of a large number of features and observations can prove highly valuable for qualitative understanding of non-linear models.

Additionally, it has been found that explicit regularization is often a key ingredient for achieving high performance of both linear and non-linear models. For example, neural networks are often trained using L2 regularization (Goodfellow et al., 2016) or dropout (Srivastava et al., 2014). By contrast, most of the above-mentioned theoretical work on linear models considers specifically L2 regularization (i.e. Ridge regression). It is quite natural, then, to think that the linear setting could be used as a test case for studying other forms of regularization. This question becomes especially interesting in light of several recent works which have explored the effect of various non-standard forms of regularization on neural networks such as rank constraints (Kamalakara et al., 2022; Yang et al., 2020) or Spectral norms (Johansson et al., 2022; Yoshida & Miyato, 2017). The Nuclear norm (or convexified rank) also plays a key role in other kinds of high-dimensional non-linear learning problems (Hu et al., 2021), most notably matrix completion, and is related to dropout regularization in neural networks (Mianjy & Arora, 2019).

Thus with the general motivation of gaining a detailed understanding of different kinds of regularizations in a tractable yet previously-validated and informative setting, we study in this work the effect of different matrix norm regularizations in the context of high-dimensional linear regression. More specifically, we consider the problem of linear regression, under the constraint that the *bias matrix* is bounded according to some fixed matrix norm. We prove a Gauss-Markov-like theorem for this setting, which characterizes the optimal estimator at a fixed level of bias, assuming the norm in question is of Schatten type. We will be specifically interested in Nuclear, Frobenius, and Spectral norms, in which case the optimal estimators have especially simple forms, which we derive.

Next, we characterize the test error of Nuclear and Spectral estimators in several random matrix ensembles (the corresponding results for Ridge regression being well known). We find an intriguing pattern: while the Ridge estimator can always attain the lowest error of any estimator, the minima can be *sharp* (as a function of regularization strength). The Nuclear estimator, by contrast, can often attain test error nearly as low as the Ridge case, but with significantly flatter minima. This has implications for practical situations when the regularization strength is selected using a noisy search procedure such as cross validation, as it could happen that such a procedure finds a better solution in the latter case, even if the actual global optimum is better in the former case. We then perform a series of simulation studies with cross-validation, in which we find that this can actually happen, with both Gaussian features and Random Fourier features, and more generally characterize the relative performance of the three estimators across a range of several generative factors. We close with a survey of related work and general discussion.

## 2 THEORY

### 2.1 SETUP AND EXTENDED GAUSS-MARKOV THEOREM FOR SCHATTEN NORMS

Throughout, we will be concerned with the classic regression problem. Denote the training data by $X$, which is an $N \times d$ matrix, and the training targets by $Y$, which is an $N \times 1$ vector. $Y$ is assumed to have distribution $Y = X\beta_0 + \epsilon$, where $\epsilon$ are independent samples from a fixed noise distribution. We will assume that the noise distribution has mean zero and variance $\sigma^2 = 1$, but otherwise do not place distributional assumptions on it. We will restrict attention to the class of *Linear Estimators*. These are models that take the form

$$\hat{\beta} = LY \tag{1}$$

Where $L$ is an $d \times N$ matrix that depends (possibly non-linearly) on the training observations $X$. Given a new datapoint $x_{test}$, the predicted value is just $\hat{y} = \langle x_{test}, \hat{\beta} \rangle$. One simple observation is that for any linear estimator the bias $\mathbb{E}_\epsilon \hat{\beta} - \beta_0$ can be expressed as a linear function of $\beta_0$:

$$\mathbb{E}_\epsilon \hat{\beta} - \beta_0 = (LX - I)\beta_0 \tag{2}$$

We thus define the bias operator $B := LX - I$. Note that the actual bias value generally depends on the unknown quantity $\beta_0$, while the variance $Var_\epsilon(\hat{\beta}) = LL^T$ does not. The classic Gauss-Markov theorem tells us that if we want to impose exactly zero bias[1], then the minimal variance estimator is the ordinary least squares estimator. But what if we want to relax this, to allow non-zero but bounded amount of bias? Taking this desideratum literally, we run into the fundamental issue that the bias depends on the unknown $\beta_0$. As a proxy, we propose to control the size of $B$ instead, using some choice of matrix norm. In what follows, we will only consider Schatten norms, $\|M\|_p = (\sum_i \sigma_i(M)^p)^{1/p}$, $p \geq 1$, although in principle other matrix norms could be used. [2]This motivates the following:

**Definition 1.** *Let $p \geq 1$, and let $C \geq 0$. For any matrix $X \in \mathbb{R}^{N \times d}$, the p-Bias constrained Linear estimator $L_p(X)$ is defined as the minimal variance estimator which has p-bias of at most C. That is,*

$$L_p(X) = argmin_{L \in \mathbb{R}^{d \times N}; \|LX - I_d\|_p \leq C} Tr(LL^T)/2 \tag{3}$$

*where $\|\cdot\|_p$ is a Schatten norm. If $Y$ is a vector of regression targets, the estimated coefficient vector is $\hat{\beta} := L_p(X)Y$.*

We now state our main theorem:

**Theorem 2.** *(Extended Gauss-Markov) Let $X \in \mathbb{R}^{N \times d}$ be a matrix of observations. Assume that $G := X^T X$ is invertible, and let $G = U diag(\sigma) U^T$ be the diagonalized form. For $p \geq 1$, the p-Bias constrained estimator with bound C can be expressed in the form $L_p(X) = \hat{G}^{-1} X^T$, where $\hat{G}$ is symmetric, simultaneously diagonalizable with G, and satisfies $G \preceq \hat{G}$[3]. For the following special cases of p we further have:*

---

[1]Note the subtle point that this is the only case in which we can exactly control the bias without knowing the value of $\beta_0$

[2]We reserve the notation $\|\cdot\|_p$ for a Schatten norm of a matrix, and use $|\cdot|_p$ for Euclidean vector norms

[3]i.e., $\hat{G} - G$ is non-negative definite

$p = 1$ *(Nuclear norm):* $\hat{G} = U diag(\max(\sigma, \alpha)) U^T$

$p = 2$ *(Frobenius norm):* $\hat{G} = G + \alpha I_d$

$p = \infty$ *(Spectral norm):* $\hat{G} = (1 + \alpha) G$

*where $\alpha \geq 0$ is determined by $C$, and $max$ denotes the elementwise maximum, i.e. $max(\sigma, \alpha)_i = max(\sigma_i, \alpha)$*

Note that the $p = 2$ case is just Ridge regression, while the $p = \infty$ case is just scalar shrinkage of $\hat{\beta}_{OLS}$ towards 0, as per Stein (1962). We refer to the $p = 1$ case as Nuclear Regression and likewise Spectral Regression for the $p = \infty$ case. The exact relation between $C$ and $\alpha$ is given in A.8.

We also note that our theorem does not perfectly generalize the classical Gauss-Markov theorem. Indeed, the classical theorem is usually stated in the form $Var(\hat{\beta}_{OLS}) \preceq Var(\hat{\beta})$ where $\hat{\beta}$ is any unbiased linear estimator (Wooldride, 2015). However, taking $C = 0$ in our Theorem 2, we would conclude only that $Tr(Var(\hat{\beta}_{OLS})) \leq Tr(Var(\hat{\beta}))$, which is strictly weaker.

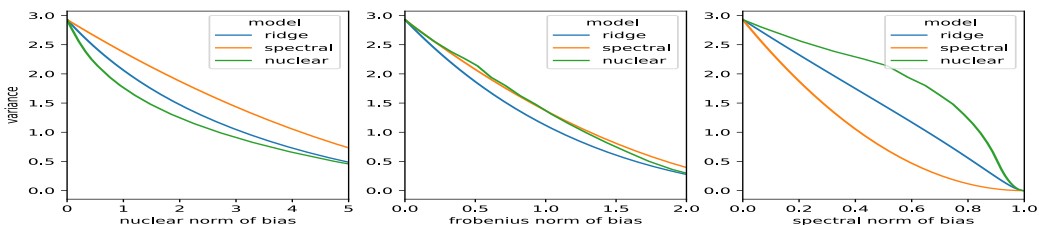

Figure 1: Illustration of Theorem 2 Each point on the curve corresponds to a regression fit with a certain value of $\alpha$, on the matrix $X = Diag(\sqrt{1}, \sqrt{2}, \ldots, \sqrt{10})$. We show the norm of the bias matrix $B = LX - I$ on the x axis, and the sample variance $Tr(L^T L)$ on the y axis. The green curve always lies below the other two on the left plot, and analogously for the other two plots.

## 2.2 FORMULAS FOR TEST ERROR

We next analytically characterize the generalization performance of the estimators derived in the previous section, under certain specific assumptions on the distribution of $X$. We will consider two different random matrix ensembles: a standard spherical Gaussian ensemble, and another with orthogonal predictors but arbitrary spectrum. We will compute the test error averaged over random datasets, in an appropriate thermodynamic limit.

### 2.2.1 SPHERICAL GAUSSIAN ENSEMBLE

Consider a predictor matrix $X \in \mathbb{R}^{N \times d}$, where each entry is an independent centered Gaussian of variance $1/N$. We consider the thermodynamic limit in which $N, d \to \infty$ and $d/N \to \lambda < 1$. The training targets are distributed as $Y_i = \langle X_i, \beta_0 \rangle + \sigma \epsilon_i$, where $\epsilon_i \sim N(0, 1)$, and the magnitude of the ground-truth coefficient vector $\beta_0$ satisfies $|\beta_0|^2/d \to \beta^2$, for some fixed $\beta > 0$. The testing error for a given dataset $X, Y$ is defined as

$$MSE := \mathbb{E}_{x \sim N(0, I_d/N)} (\langle x, \beta_0 \rangle - \langle x, \hat{\beta} \rangle)^2, \qquad (4)$$

where $\hat{\beta}$ is the coefficient vector estimated from $X$ and $Y$. (Note that we do not include exogenous noise in the testing data; if we did, it would just contribute an additive offset of $\sigma^2$). We observe that the expression can be simplified to $|\beta_0 - \hat{\beta}|^2/N$, however we write it in the more verbose form above to make clear that it is obtained as an average over the distribution of the testing point $x$.

We are interested in the average value of this over different datasets. Using standard techniques in Random Matrix theory, it is not hard to show (cf. A.6):

**Proposition 2.1.** *Consider the p-bias constrained estimator with regularization strength $\alpha$.*

*In the above limit, the average test error $Err_p(\alpha) := \mathbb{E}_{X,Y} MSE$ is given by*

$$Err(\alpha) = \lambda \int \beta^2 (1 - \frac{x}{f_\alpha(x)})^2 + \sigma^2 \frac{x}{f_\alpha(x)^2} \mu_{MP}^\lambda(dx) \tag{5}$$

*where $\mu_{MP}^\lambda$ is the Marchenko-Pastur density with concentration parameter $\lambda$, and $f_\alpha$ is defined casewise as follows:*

$$f_\alpha(x) = max(x, \alpha), p = 1 \tag{6}$$
$$f_\alpha(x) = x + \alpha, p = 2 \tag{7}$$
$$f_\alpha(x) = x(1 + \alpha), p = \infty \tag{8}$$

In practice, we use this formula and evaluate the integral using numerical quadrature. However, it is interesting to note that the integral can be actually be further simplified in each of the three cases. For the ridge case, there is a well-known formula in terms of the Stieltjes transform of $\mu_{MP}^\lambda$ that easily follows from the above integral(Bai & Silverstein, 2010). In the spectral case, we can derive a formula using simple calculus:

**Proposition 2.2.** *For the Spectral estimator in the spherical Gaussian setup, the test error in the thermodynamic limit equals*

$$Err_\infty(\alpha) = \frac{\lambda\beta^2\alpha^2 + \frac{\lambda}{1-\lambda}\sigma^2}{(1+\alpha)^2} \tag{9}$$

To state the result for the Nuclear case,it is necessary to introduce the Appel hypergeometric function $F_1$(Appell, 1925), which generalizes the classical Gauss hypergeometric function to two variables.

**Proposition 2.3.** *For the Nuclear estimator in the spherical Gaussian setup, the test error in the thermodynamic limit can be expressed in terms of the Appel hypergeometric function $F_1$:*

$$Err_1(\alpha) = \begin{cases} \frac{\sigma^2\lambda}{1-\lambda} & \alpha \leq \lambda_- \\ \frac{\sigma^2\lambda}{1-\lambda} + \lambda\beta^2 CDF_\lambda(\alpha) + \sqrt{\alpha - \lambda_-} \sum_{r \in \{-1,1,2\}} c_r(\alpha) F_1(\frac{3}{2}, 1-r, -\frac{1}{2}, \frac{5}{2}, 1 - \frac{\alpha}{\lambda_-}, \frac{\alpha-\lambda_-}{\lambda_+-\lambda_-}) & \lambda_- \leq \alpha \leq \lambda_+ \\ \frac{\lambda(1+\lambda)}{\alpha^2} + \frac{\lambda\sigma^2 - 2\lambda\alpha\beta^2}{\alpha^2} + \lambda\beta^2 & \alpha \geq \lambda_+ \end{cases}$$

*where $\lambda_\pm$ are the limits of the Marchenko-Pastur support: $\lambda_\pm = (1\pm\sqrt{\lambda})^2$, $CDF_\lambda$ is the cumulative distribution function of $\mu_{MP}$ and $c_r(\alpha)$ are certain rational functions of $\alpha$.*

See A.5 for proof, and precise definition of $F_1$.

### 2.2.2 DIAGONAL ENSEMBLE

While the spherical Gaussian case is an instructive starting point, the assumptions are clearly somewhat limiting. In particular, it implies a very specific functional form of the spectral density, which may not be a good match to real data. Thus, we analyze another random matrix ensemble which allows us the flexibility to specify an essentially arbitrary spectral density. On the one hand, this new ensemble has its own limiting assumptions that the Gram matrix $X^T X$ is almost surely diagonal, and the observations (i.e. rows of $X$) are no longer independent. But on the other, having another random matrix model allows us to assess whether qualitative observations about the spherical Gaussian case generalize to other settings.

We now describe the random matrix ensemble. We first specify some distribution $\nu$ supported on $[0, 1]$, which will be the spectral density. We also specify some distribution $\nu_{noise}$ supported on a bounded interval of $\mathbb{R}_+$, which will act as multiplicative random noise on the spectrum; we require that $\mathbb{E}_{x \sim \nu_{noise}} x = 1$. To generate training and testing data, we first sample $\lambda_i \sim \nu, i = 1, \ldots, d$ independently, and $s_i \sim \nu_{noise}$. We then set

$$X_{tr} = X_1 diag(\{\sqrt{\lambda_i s_i}\}_i), X_{test} = X_2 diag(\{\sqrt{\lambda_i}\}_i) \tag{10}$$

where $X_1$ and $X_2$ are drawn independently and uniformly from the Stiefel manifold $O(n, d)$ of $N \times d$ orthogonal matrices. That is $X_i^T X_i = I_d$. We observe that, as noted above: 1.) the Gram matrix $G = X_{tr}^T X_{tr}$ is diagonal almost surely, and 2.) the limiting eigenvalue density of $G$ is given by $\nu$.

The regression targets are then constructed as before: $Y_{tr} = X_{tr}\beta_0 + \epsilon, Y_{test} = X_{test}\beta_0$. The test error is defined as

$$MSE := N^{-1}|X_{test}\hat{\beta} - X_{test}\beta_0|^2 \tag{11}$$

Note that this is slightly different from the expression for MSE in the spherical Gaussian case (Equation 4). The basic reason is that the observations (i.e. rows) are here not independent. Thus, in contrast to the spherical Gaussian case, where we could sample one test observation at a time and average, here we sample a batch of N (dependent) test observations at a time, and compute the average error over those N. As before, we will be interested in the value of this error, marginalized over all of the randomness, and taken in the thermodynamic limit.

**Proposition 2.4.** *In the thermodynamic limit of the above random matrix ensemble, the average test error $Err_p(\alpha) = \mathbb{E}_{X_{tr}, X_{test}, \epsilon, s} MSE$ can be expressed as*

$$Err_p(\alpha) = \lambda \int_0^1 \beta^2 x (1 - \frac{x}{f_\alpha(x)})^2 + \sigma^2 \frac{x^2}{f_\alpha(x)^2} \nu(dx) \tag{12}$$

*where $f_\alpha$ has the same form as in Proposition 2.1.*

See A.6 for proof. Note the similarity to the result of Proposition 2.1. The basic reason for this is that both ensembles have a similar rotational invariance, which allows the MSE to be expressed as a sum over the spectrum of $X_{tr}$. Besides the integrating measure, the only difference is an extra factor of $x$ in the Diagonal case.

In our experiments, we specialize to power law distributions $d\nu(x)/dx \propto x^{\gamma-1}\mathbf{1}_{x \in [0,1]}$, as many real covariance matrices have approximately this structure (Stringer et al., 2019; Qin & Colwell, 2018; Liu et al., 1997; Ruderman & Bialek, 1994). In this case, the expressions for the test error can again be simplified to analytic expressions that do not involve integrals, see Section A.9 for details.

### 2.2.3 ON THE OPTIMAL REGULARIZATION FORM

It is not hard to see that the derivations in the above two sections go through for any estimator of the form $\hat{\beta} = \hat{G}^{-1}X^TY$, where the eigenvalues of $\hat{G}$ are related to those of $G$ through some fixed point-wise transform $\lambda_i(\hat{G}) = f(\lambda_i(G))$, and $\hat{G}$ is simultaneously diagonalizable with $G$. It is thus natural to ask what would be optimal estimator in this class, *if we are allowed to select f arbitrarily*? By formally differentiating the integrand with respect to $f$ and setting equal to zero[4], it is easy to see that in both of the above random matrix models, the optimal form of $f$ corresponds to Ridge regression with regularization strength $\sigma^2/\beta^2$.

Does this mean that Ridge regression will always perform better in practice than any other estimator? Not necessarily. For the above estimator is an *oracle estimator* in the sense that it requires knowledge of the ground-truth $\beta$ and $\sigma$, which are typically not known. In practice, we would estimate the optimal ridge parameter using,e.g., cross-validation, and hope that our estimate is close to the oracle-optimal value $\sigma^2/\beta^2$. In the finite-sample regime, there will be some noise in the cross-validated errors, so this essentially amounts to trying to minimize $Err(\alpha)$ given only access to a small number of noisy estimates of the underlying function. Additionally, there will be "resolution noise" caused by the fact that we will typically only try to estimate the test error for a small number of values of $\alpha$. A very rough rule of thumb is that if we sample $n$ values $\alpha_i$, all within some small distance $\delta$ to the minimum, then $min_i Err(\alpha_i) \approx \mu + \frac{(\kappa\delta)^2}{(n+1)(n+2)}$, where $\mu = \min_\alpha Err(\alpha)$ is the true minimum, and $\kappa^2$ is the Hessian of $Err(\alpha)$ at the minimum (see A.3 for justification). We conclude that the "effective minimum" that we would find using cross validation is penalized by higher values of curvature, especially for small $n$.

### 2.2.4 EXPERIMENTAL VALIDATION

In order to validate the above formulas, we generated synthetic datasets according to the random matrix ensembles from sections 2.2.1 and 2.2.2 , and compared the average test error with the theoretically predicted values. For each such dataset, we used $N = 100$ training observations, and obtained the empirical test error by averaging over 100 datasets. We did this for both the spherical

---

[4]this can be justified using results from Calculus of Variations

Gaussian case, as well as the diagonal case with a power law spectral density $d\nu/dx \propto x^{\gamma-1}\mathbf{1}_{x\in[0,1]}$. In Figure 2 we show several plots with different values of the other hyperparameters; in all cases we see a very close correspondence between the empirical and predicted errors. In accordance with the discussion in the previous section, we can also see visually that the minima of the Nuclear estimator are often noticeably flatter than those of the Ridge, while also being only moderately higher.

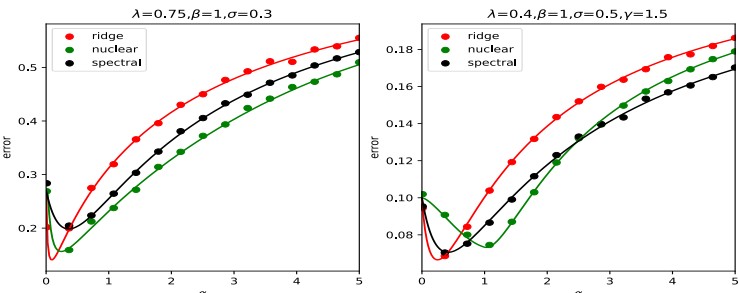

Figure 2: Predicted vs. empirical test errors, for the spherical Gaussian case (left) and diagonal case with power law spectrum (right). Each dot represents an average over 20 different datasets, each with $N = 100$ training observations. Note that the minima for the Ridge curve are notably steeper than for the other two.

## 3 EXPERIMENTS

### 3.1 LOSS BASIN GEOMETRY COMPARISON

Here, we aim to systematize the observation that the Nuclear estimator can have slightly higher, but much flatter, minima than the Ridge estimator. To do so, we simply fix the hyperparameters $\sigma, \lambda$, and consider the theoretical test error as a function of $\alpha$. We record both the minimal value over $\alpha$ $m := \min_\alpha Err(\alpha)$ as well as the curvature at the minimum: $\kappa := \sqrt{\partial^2 Err(\alpha)/\partial\alpha^2}|_{\alpha=\alpha_{min}}$. For all cases, we fix $\beta = 1$. For the sake of having an interpretable frame of reference, we report these values as percentage increases relative to the corresponding values for the Ridge estimator, shown in Figure 3. For the nuclear case, we can plainly see that the minima of the test loss are nearly as deep as for the the Ridge (within a few percentage points), however the minima are often very substantially flatter (corresponding to negative values in the table). The Spectral estimator tends to also have flatter minima than Ridge, but with a less favorable depth tradeoff compared to the Nuclear. Moreover, we see a pronounced effect of $\sigma$, with increasing values tending to decrease the gap between the depths of Nuclear and Ridge minima, while also decreasing the difference in curvature. As in Figure 4, we see that these observations largely carry over to the case of a power-law spectral density, indicating that they are not simply an incidental property of the spherical Gaussian model. See A.7 for more technical details on the calculation of these values.

min/curvature at min, relative to Ridge

| | nuclear | | | | | spectral | | | |
| --- | --- | --- | --- | --- | --- | --- | --- | --- | --- |
| | λ=0.2 | λ=0.4 | λ=0.6 | λ=0.8 | | λ=0.2 | λ=0.4 | λ=0.6 | λ=0.8 |
| σ=0.5 | 12.3/-49.8 | 12.7/-51.3 | 11.7/-53.5 | 10.0/-53.8 | σ=0.5 | 4.3/-18.8 | 12.3/-33.5 | 27.8/-49.2 | 61.2/-69.3 |
| σ=1 | 3.9/-12.3 | 5.9/-39.3 | 6.3/-46.1 | 6.1/-49.0 | σ=1 | 5.8/-15.3 | 13.7/-33.7 | 24.6/-54.2 | 39.8/-78.9 |
| σ=1.5 | 0.7/-7.2 | 1.2/-6.0 | 1.6/-14.7 | 1.8/-14.3 | σ=1.5 | 4.6/-17.8 | 10.1/-39.3 | 16.5/-61.4 | 24.2/-84.2 |
| σ=2 | 0.2/-0.3 | 0.3/-2.0 | 0.4/-3.9 | 0.5/-5.8 | σ=2 | 3.4/-20.7 | 7.1/-44.1 | 11.2/-66.3 | 15.7/-85.6 |
| σ=2.5 | 0.1/-3.0 | 0.1/3.9 | 0.1/-0.3 | 0.2/-4.5 | σ=2.5 | 2.5/-22.7 | 5.1/-47.0 | 7.9/-68.9 | 10.8/-87.1 |

Figure 3: Quantification of loss basin geometry for the spherical Gaussian model. Each entry in the table shows the minimal attainable test error/curvature at the minimum, where each is expressed as a percentage increase relative to the corresponding value for the Ridge estimator.

| nuclear,γ=0.5 | λ=0.2 | λ=0.4 | λ=0.6 | λ=0.8 | spectral,γ=0.5 | λ=0.2 | λ=0.4 | λ=0.6 | λ=0.8 |
|---|---|---|---|---|---|---|---|---|---|
| σ=0.25 | 12.3/-70.7 | 12.3/-70.7 | 12.3/-70.7 | 12.3/-70.7 | σ=0.25 | 26.0/-62.8 | 26.0/-62.8 | 26.0/-62.8 | 26.0/-62.8 |
| σ=0.5 | 12.3/-58.5 | 12.3/-58.5 | 12.3/-58.5 | 12.3/-58.5 | σ=0.5 | 28.0/-57.6 | 28.0/-57.6 | 28.0/-57.6 | 28.0/-57.6 |
| σ=0.75 | 3.1/-8.5 | 3.1/-8.5 | 3.1/-8.5 | 3.1/-8.5 | σ=0.75 | 22.2/-60.4 | 22.2/-60.4 | 22.2/-60.4 | 22.2/-60.4 |
| σ=1 | 1.0/-7.0 | 1.0/-7.0 | 1.0/-7.0 | 1.0/-7.0 | σ=1 | 16.5/-66.2 | 16.5/-66.2 | 16.5/-66.2 | 16.5/-66.2 |

| nuclear,γ=1 | λ=0.2 | λ=0.4 | λ=0.6 | λ=0.8 | spectral,γ=1 | λ=0.2 | λ=0.4 | λ=0.6 | λ=0.8 |
|---|---|---|---|---|---|---|---|---|---|
| σ=0.25 | 11.4/-76.1 | 11.4/-76.1 | 11.4/-76.1 | 11.4/-76.1 | σ=0.25 | 8.0/-51.0 | 8.0/-51.0 | 8.0/-51.0 | 8.0/-51.0 |
| σ=0.5 | 11.5/-42.0 | 11.5/-42.0 | 11.5/-42.0 | 11.5/-42.0 | σ=0.5 | 11.5/-43.4 | 11.5/-43.4 | 11.5/-43.4 | 11.5/-43.4 |
| σ=0.75 | 2.9/1.7 | 2.9/1.7 | 2.9/1.7 | 2.9/1.7 | σ=0.75 | 10.6/-43.3 | 10.6/-43.3 | 10.6/-43.3 | 10.6/-43.3 |
| σ=1 | 0.9/0.2 | 0.9/0.2 | 0.9/0.2 | 0.9/0.2 | σ=1 | 8.6/-48.5 | 8.6/-48.5 | 8.6/-48.5 | 8.6/-48.5 |

| nuclear,γ=1.5 | λ=0.2 | λ=0.4 | λ=0.6 | λ=0.8 | spectral,γ=1.5 | λ=0.2 | λ=0.4 | λ=0.6 | λ=0.8 |
|---|---|---|---|---|---|---|---|---|---|
| σ=0.25 | 10.4/-80.5 | 10.4/-80.5 | 10.4/-80.5 | 10.4/-80.5 | σ=0.25 | 3.5/-42.3 | 3.5/-42.3 | 3.5/-42.3 | 3.5/-42.3 |
| σ=0.5 | 10.1/-30.7 | 10.1/-30.7 | 10.1/-30.7 | 10.1/-30.7 | σ=0.5 | 6.1/-35.2 | 6.1/-35.2 | 6.1/-35.2 | 6.1/-35.2 |
| σ=0.75 | 2.5/-1.1 | 2.5/-1.1 | 2.5/-1.1 | 2.5/-1.1 | σ=0.75 | 6.2/-33.8 | 6.2/-33.8 | 6.2/-33.8 | 6.2/-33.8 |
| σ=1 | 0.8/-5.0 | 0.8/-5.0 | 0.8/-5.0 | 0.8/-5.0 | σ=1 | 5.3/-38.5 | 5.3/-38.5 | 5.3/-38.5 | 5.3/-38.5 |

| nuclear,γ=2 | λ=0.2 | λ=0.4 | λ=0.6 | λ=0.8 | spectral,γ=2 | λ=0.2 | λ=0.4 | λ=0.6 | λ=0.8 |
|---|---|---|---|---|---|---|---|---|---|
| σ=0.25 | 9.7/-83.9 | 9.7/-83.9 | 9.7/-83.9 | 9.7/-83.9 | σ=0.25 | 1.9/-35.7 | 1.9/-35.7 | 1.9/-35.7 | 1.9/-35.7 |
| σ=0.5 | 8.7/-25.2 | 8.7/-25.2 | 8.7/-25.2 | 8.7/-25.2 | σ=0.5 | 3.7/-31.5 | 3.7/-31.5 | 3.7/-31.5 | 3.7/-31.5 |
| σ=0.75 | 2.2/-5.6 | 2.2/-5.6 | 2.2/-5.6 | 2.2/-5.6 | σ=0.75 | 4.0/-30.9 | 4.0/-30.9 | 4.0/-30.9 | 4.0/-30.9 |
| σ=1 | 0.7/-0.7 | 0.7/-0.7 | 0.7/-0.7 | 0.7/-0.7 | σ=1 | 3.5/-31.6 | 3.5/-31.6 | 3.5/-31.6 | 3.5/-31.6 |

Figure 4: Same as Figure 3 except for the Diagonal matrix model with power law spectral density.

## 3.2 SIMULATION STUDIES

The purpose of this section is to show that the observations about the loss basin geometry can have practical consequences in settings where the regularization strength is selected by cross-validation.

### 3.2.1 GAUSSIAN PREDICTORS

We generated training data by sampling $X_i \sim N(0, C(\rho))$, where $C(\rho) := (1 - \rho)I_d + \rho\mathbf{1}\mathbf{1}^t$. We then generated a ground truth coefficient vector by sampling $\beta_0 \sim N(0, I_d)$, and training targets $y_i$ by $y_i \sim N(\langle X_i, \beta_0 \rangle, \sigma^2)$. The testing data were generated similarly, using the same $\beta_0$. In all cases, we used $N = 100$ training examples, and 5000 testing examples. For each combination of hyperparameters $d, \sigma, \rho$, we generated 100 train/test datasets. For each model (Spectral, Ridge, Nuclear) and each individual dataset, we used 3-fold cross validation to select the best-performing $\alpha$ on the training set. We then refit the model on the entire training set, and evaluated its performance on the test set. The set of allowable $\alpha$ values considered in the cross-validation was the same for all models, and consisted of 9 equally logarithmically spaced values spanning $10^{-4}$ to $10^{6}$.

In Figure 5 we plot the best-performing model for each combination of hyperparameters, for two different definitions of "best". In the top row, we show the model that attains the lowest average test error in each cell. In the bottom, we show the model that is most likely to "win" on any given dataset. That is, suppose we evaluate model $m$ on dataset $i$, and obtain a test error of $MSE_{mi}$. The top row shows $argmin_m \mathbb{E}_j MSE_{mj}$, while the bottom row shows $Mode(\{argmin_m MSE_{mj}\}_j)$. We see that in terms of average error, the Ridge and Nuclear are essentially matched, with Spectral being a distant 3rd. The Nuclear however is overwhelmingly likely to be the winner on a given dataset.

### 3.2.2 REGRESSION OF NONLINEAR FUNCTIONS USING RANDOM FOURIER FEATURES

In order to generalize our observations beyond the Gaussian features case, we consider a more complex setup in which the true relationship between the predictors and the targets is non-linear, and the regression is performed in the feature space of a universal approximating kernel. To be more specific, the process for generating data was as follows. We first pick integers $d, d_{rbf}$ which

**d=20,best mse**

| | σ=0.5 | σ=1.5 | σ=2.0 | σ=2.5 | σ=3.0 | σ=3.5 |
|---|---|---|---|---|---|---|
| ρ=0.0 | N | R | R | S | S | S |
| ρ=0.25 | R | R | N | R | S | S |
| ρ=0.5 | R | R | N | R | R | S |
| ρ=0.75 | R | R | R | R | S | S |
| ρ=0.9 | N | N | N | R | N | N |

**d=40,best mse**

| | σ=0.5 | σ=1.5 | σ=2.0 | σ=2.5 | σ=3.0 | σ=3.5 |
|---|---|---|---|---|---|---|
| ρ=0.0 | N | R | S | S | S | S |
| ρ=0.25 | R | N | R | R | R | N |
| ρ=0.5 | R | R | N | N | N | R |
| ρ=0.75 | N | R | R | R | R | R |
| ρ=0.9 | R | N | N | N | N | N |

**d=60,best mse**

| | σ=0.5 | σ=1.5 | σ=2.0 | σ=2.5 | σ=3.0 | σ=3.5 |
|---|---|---|---|---|---|---|
| ρ=0.0 | R | R | R | S | S | S |
| ρ=0.25 | R | R | N | N | R | N |
| ρ=0.5 | N | R | R | R | N | N |
| ρ=0.75 | R | R | R | R | R | N |
| ρ=0.9 | R | N | N | N | R | N |

**d=80,best mse**

| | σ=0.5 | σ=1.5 | σ=2.0 | σ=2.5 | σ=3.0 | σ=3.5 |
|---|---|---|---|---|---|---|
| ρ=0.0 | R | R | R | S | S | S |
| ρ=0.25 | N | R | N | N | N | N |
| ρ=0.5 | N | R | R | R | R | R |
| ρ=0.75 | R | N | R | N | R | N |
| ρ=0.9 | R | N | N | N | N | N |

**d=20, prob best**

| | σ=0.5 | σ=1.5 | σ=2.0 | σ=2.5 | σ=3.0 | σ=3.5 |
|---|---|---|---|---|---|---|
| ρ=0.0 | N | N | N | S | N | N |
| ρ=0.25 | R | R | N | R | N | S |
| ρ=0.5 | R | N | N | N | S | N |
| ρ=0.75 | R | S | N | N | N | S |
| ρ=0.9 | N | N | N | R | N | S |

**d=40, prob best**

| | σ=0.5 | σ=1.5 | σ=2.0 | σ=2.5 | σ=3.0 | σ=3.5 |
|---|---|---|---|---|---|---|
| ρ=0.0 | N | R | N | S | N | N |
| ρ=0.25 | R | N | N | N | S | N |
| ρ=0.5 | N | N | N | N | N | N |
| ρ=0.75 | N | R | N | N | S | N |
| ρ=0.9 | N | N | N | R | R | N |

**d=60, prob best**

| | σ=0.5 | σ=1.5 | σ=2.0 | σ=2.5 | σ=3.0 | σ=3.5 |
|---|---|---|---|---|---|---|
| ρ=0.0 | N | R | S | N | N | N |
| ρ=0.25 | R | N | N | S | S | S |
| ρ=0.5 | N | N | N | N | N | N |
| ρ=0.75 | N | R | N | N | N | N |
| ρ=0.9 | N | N | N | N | N | N |

**d=80, prob best**

| | σ=0.5 | σ=1.5 | σ=2.0 | σ=2.5 | σ=3.0 | σ=3.5 |
|---|---|---|---|---|---|---|
| ρ=0.0 | R | R | N | N | N | N |
| ρ=0.25 | N | N | N | N | N | N |
| ρ=0.5 | N | R | N | N | N | N |
| ρ=0.75 | N | R | R | N | R | N |
| ρ=0.9 | R | N | N | N | R | N |

Figure 5: Best performing models with cross-validation on Gaussian predictors. Top: best average error. Bottom: highest probability of winning. Each cell is an aggregate over 100 datasets.

define the dimension of the observed space, and of the kernel feature map, respectively. We sampled a training matrix $X$ of size $N \times d$, in which each entry has variance $1/d_{rbf}$. We then sampled a non-linear function $f : \mathbb{R}^d \to \mathbb{R}$ in a manner described below, and generated training targets as $y_i = f(X_i) + \sigma \epsilon_i$, where $X_i$ is the ith row of $X$ (i.e. ith observation) and $\epsilon_i$ is unit Gaussian. Testing data was generated similarly, using the very same function $f$ (and setting $\sigma = 0$ as in the previous analyses). Since the relation between $X$ and $Y$ is non-linear, we do not directly fit the models on $X$, but rather first fit a Random Fourier features (RFF) model (Rahimi & Recht, 2008) on $X$, using $d_{rbf}$ features [5]. We then fit the linear models on $RFF(X), Y$, where $RFF(X)$ is the $N \times d_{rbf}$ matrix obtained by applying the Random Fourier features mapping. To estimate test error, we first transform $X_{test}$ using the *same* RFF mapping, and consider $|RFF(X_{test})\hat{\beta} - Y_{test}|^2$. Finally, the non-linear function $f$ was defined as follows:

$$f(x) = \sum_{k=1}^{d_{rbf}} cos(2\pi k \langle x, v_i \rangle)/k^2 \tag{13}$$

where $v_i$ are sampled uniformly from the unit sphere $S^{d-1} \subset \mathbb{R}^d$. We fixed $d = 10$ and $N = 100$ in all cases. For each value of $d_{rbf}$ and $\sigma$, we generated 100 datasets in this way. We did not consider the Spectral estimator for this experiment, because it does not naturally accomodate the overparametrized case. As in Figure 6, we see that the cross-validated Nuclear estimator can again outperform the cross-validated Ridge, with the effect becoming more pronounced for larger values of $\sigma$.

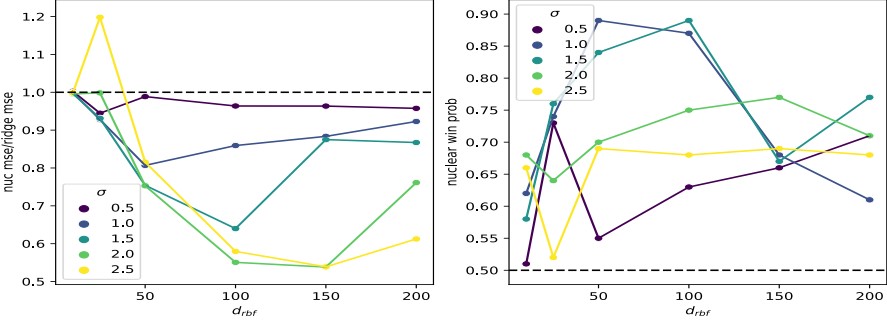

Figure 6: Test set performance for cross-validated Nuclear estimator on the Random Fourier features data. Left shows average test error expressed as a ratio relative to average Ridge test error. Right shows probability of "winning" on a given dataset. Each dot represents an aggregate over 100 datasets.

---

[5]The RFF fit was done using the class `sklearn.kernel_approximation.RBFSampler`.

## 4 RELATED WORK

Several other sorts of generalized Gauss-Markov theorems have appeared in the literature, although they generalize along different axes than our formulation, such as allowing non-spherical errors (Kourouklis & Paige, 1981), predictor collinearity (Lewis & Odell, 1966), random effects (Shaffer, 1991), or non-linear estimators (Hansen, 2022). Similarly,the Nuclear and Spectral estimators bear a very strong resemblance to various notions from classical statistics, most notably Principal Components Regression (PCR) (Kendall, 1957), Stein Shrinkage (Stein, 1962) respectively. Finally, the work of Hocking et al. (1976) is particularly relevant and worthy of further discussion here-we give a more detailed comparison in A.2.

Lately, there there has also been renewed interest in the study of high-dimensional regression models and the effects of various kinds of regularization thereupon. Most such works focus on Ridge or kernel Ridge regression ((Mei & Montanari, 2020; Dobriban & Wager, 2018; Canatar et al., 2021; Dicker, 2016; Belkin et al., 2020; Hu & Lu, 2022; Nakkiran et al., 2021)). Ridge regression has also proven a powerful lens to understand other techniques such as dropout (Wager et al., 2013), data augmentation(Lin et al., 2023) and early stopping (Ali et al., 2019). This pervasiveness of the ridge underscores our claim that a detailed understanding of other kinds of regularization than L2 in the linear case could be used to study more complex regularizations in non-linear settings.

Some other work has analyzed other regularizations than ridge such as Bayati & Montanari (2011); Samet et al. (2013)for Lasso. Theoretical properties of nuclear norm have mainly been studied in the context of matrix completion, in which it is often used as a surrogate for matrix rank (Candès & Recht, 2012; Koltchinskii et al., 2011), but see Hu et al. (2021) for other applications. Another setting in which rank penalties appear in regression problems is Low Rank Regression (Bunea et al., 2011; Izenman, 2008), although this is fairly different than our setting as it has to do with regularizing the *outputs* of a regression problem with multiple dependent variables.There is also some work on theoretical properties of the Principal Components Regression estimator (Xu & Hsu, 2019); optimizing the number of PCR components has a similar flavor to optimizing $\alpha$ in the Nuclear estimator.

Finally, flat minima are also popular as an explanatory tool in the context of generalization of neural networks (Baldassi et al., 2021; Mulayoff & Michaeli, 2020; Hochreiter & Schmidhuber, 1997), in which the randomness in the minimization procedure is induced by stochastic gradient descent.

## 5 DISCUSSION

We have introduced two simple Linear models:Spectral and Nuclear regression, and have shown how the forms of both of these models, together with Ridge regression, may all be derived from an extension of Gauss-Markov Theorem for Schatten norm constraints on the bias operator. By considering theoretical error curves as a function of regularization strength, we observed in multiple different random matrix ensembles that the Nuclear model can often attain minima that are much flatter than those of the Ridge model, while being nearly as deep. Finally, we showed that this tradeoff can have practical consequences in a cross-validation setting, for both Gaussian and non-Gaussian features.

It should be noted that the particulars of the cross-validation results depend on the number of $\alpha$ values used in the grid search, among other factors; and as noted before, the effect of the curvature at the minimum decreases as more values of $\alpha$ are used. However, increasing the number of $\alpha$ values has other potential costs such as risk of overfitting or increased computational cost. But more fundamentally,our intention with the simulation results is not to claim that the Nuclear estimator should always be preferred over the Ridge, but rather to show that it *can* outperform in some practically relevant setups, and to use our theoretical analysis to give a precise characterization of why this happens.

There are many potential future directions of this work, such as further understanding of the Spectral and Nuclear regularizations in the kernelized or overparametrized regime, or using insights from the Nuclear case to understand other situations that involve rank constraints such as matrix completion. More broadly, we hope that this framework will provide a test case to understand less common regularizations such as $\|\cdot\|_1$ and $\|\cdot\|_\infty$, with the goal of transferring the insights gained thereby to the study of high-dimensional non-linear models.

ACKNOWLEDGMENTS

SS is supported by a T32 training grant in Computational Neuroscience (T32MH065214).

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
