# OpenReview forum: "Flat Minima in Linear Estimation and an Extended Gauss Markov Theorem"
_ICLR.cc/2024/Conference — ICLR 2024 poster_

### Official Review · Reviewer_v15j · 2023-10-19

**Soundness:** 3 good
**Presentation:** 2 fair
**Contribution:** 3 good
**Rating:** 6
**Confidence:** 4

**Summary:**

This paper considers the trade-off between the variance (MSE) and bias (bias norm) for the regularized linear regression model, where the authors consider various Schatten norms for the regularized term. Then, the authors derive the expected MSE for the Gaussian and diagonal ensembles and compare it with the ridge regression model's one and show that the ridge regression model is not always the best in term of the trade-off by using some experiments on sythetic datasets.  There exit some similar works such as Bayatti and Montanari (2011), Samet et al. (2013) for Lasso, however the authors limit their work to the class of Schatten norms.

**Strengths:**

+  The authors can obtain an exact expression for the average test error (MSE) for the spherical Gaussian model, and this bound is nearly matched to the experiment results (cf. Figure 2).
+ Experiments show that Ridge regression, which used Frobenius norm, is not always the best option for the linear model in term of the trade-off between the variance (MSE) and the bias (the bias norm). More specifically, the authors show that the Frobenius norm and the nuclear one are likely to have the same average MSE on the spherical Gaussian ensemble (model) or the diagonal matrix model, but using the nuclear norm usually achieves better MSE than the Frobenius norm. This fact also holds when mapping the dataset via a random fourier transform (RFF).

**Weaknesses:**

+ The theoretical results only hold in the thermodynamic limit $N\rightarrow \infty, d \rightarrow \infty$ and $d/N\rightarrow \lambda$. This means that the results only hold when the number of observations is linear to the signal dimension. However, in common high-dimensional settings, the number of observations is usually sub-linear to the signal dimension.
+ The result looks not an extension of the Gauss-Markov theorem since it only holds under expectation over $X$ when $X$ is an Gaussian ensemble or a diagonal ensemble. The Gauss-Markov theorem works for any $X$.
+ It looks more interesting to compare your experiment results with other norms (outside the class of  Schatten norms) such as between the nuclear norm and Lasso (norm-$1$).
+ Too many typos. Please check and correct them.

**Questions:**

How do your results in comparison with other norms which don't belong to Schatten class of norms such as Lasso (norm-$1$)?

Besides, please check and correct the following typos and unprecise.

+ $L \in \mathbb{R}^{k \times N} \rightarrow L \in \mathbb{R}^{d \times N}$
+ p.2, line 21 from the top: $\mbox{var}\_{\epsilon}=L^T L$ should be changed to $\mbox{var}\_{\epsilon}=\sigma^2 L L^T$. To keep the later, you may change the definition of $\hat{\beta}$ to $L^T(X)Y$ throughout your paper.
+ In the definition 1, you aim to minimize the variance subject to a constraint on bias by $C$. But, in the later (Theorem 2, Figure 1, etc.), it seems to me that you don't mention $C$ again, but only mention $\alpha$. At least you should mention what is $\alpha$ as a function of $C$ or vice versa in Theorem 2.
+ In Figure 1, you only plot for the very special case when $X$ is diagonal. You should also plot for other cases of $X$.
+ Typo in Theorem 2.1, the expectation should be over $\epsilon$ only since you already take expectation over $X$ in MSE, and $Y$ is a function of $X$ and $\epsilon$.
+ Right below Figure 2: Figure 3.1 $\rightarrow$ Figure 3.
+ Similarly, Figure 3.2.1 $\rightarrow$ Figure 5.

---

> ### Author Response · Authors · 2023-11-13
> **Response to Reviewer v15j**
>
> Thank you for your helpful comments. Below we provide responses to specific points.
>
> >The theoretical results only hold in the thermodynamic limit . This means that the results only hold when the number of observations is linear to the signal dimension. However, in common high-dimensional settings, the number of observations is usually sub-linear to the signal dimension.
>
> This is an interesting observation. If you have a specific paper or reference in mind here, we would be happy to cite it and discuss this point as a limitation in the Discussion section.
>
> >The result looks not an extension of the Gauss-Markov theorem since it only holds under expectation over $X$ when X is an Gaussian ensemble or a diagonal ensemble. The Gauss-Markov theorem works for any X.
>
> We apologize for the lack of clarity here. What we refer to as the Extended Gauss-Markov Theorem is actually Theorem 2 (and we have now clearly labeled this in the text in the statement of the theorem). You are correct that the results about the test error in sections 2.2.1 and 2.2.2 only hold under specific distributional hypotheses. In contrast, Theorem 2 does not make any distributional assumptions, or indeed have any probabalistic component whatsoever- it is purely a statement about an optimization problem which holds for basically arbitrary X (technically X must be full rank). However, if we assume that errors epsilon_i are uncorrelated, zero mean, and have finite variance, then we have the probabilistic interpretation that LL^T is the covariance matrix and LX-I is the bias operator. So while this interpretation does impose some distributional assumptions, it is not any more than found in the classical Gauss-Markov theorem (Note that, in the original submission  we required Gaussian error distributions, but as a consequence of this discussion have realized this is not actually necessary).
>
> >It looks more interesting to compare your experiment results with other norms (outside the class of Schatten norms) such as between the nuclear norm and Lasso (norm-1)
>
> We have done so and included results in the appendix (section A.10). The summary is that the spectral and nuclear models are competitive with lasso even when the ground truth data has sparse structure.
>
> It is also worth noting here that the Lasso is not manifestly a “Linear estimator” in the sense we define it in Section 2, even allowing for the use of norms outside the Schatten family. As a reminder, in order to qualify as a Linear estimator in our sense, the coefficients must take the form LY, where L depends only on X. It is not at all obvious that this is the case for Lasso, although we do not have a proof to the contrary.
>
> *Edit:*Actually,it is easy to prove this about Lasso. Indeed, in the case of $d=1$ (i.e. one predictor variable),the Lasso with regularization $\alpha$ has the well-known solution $w=sign(\langle X,Y\rangle)max(\langle X,Y\rangle/|X|^2-\alpha,0)$,which is clearly not a linear function of $Y$.
>
> >Too many typos. Please check and correct them.
>
> We have corrected a large number of small typos, see the general response to reviewers as well as the individual responses to the other reviewers.
>
> >How do your results in comparison with other norms which don't belong to Schatten class of norms such as Lasso (norm-1)?
>
> See above
>
> >k\times N\to d\times N
>
> Corrected
>
> >p.2, line 21 from the top: var =L^TL should be changed to sigma^2 LL^T . To keep the later, you may change the definition of beta throughout your paper.
>
> We have corrected the variance from L^TL to LL^T in both the main text and supplement. As far as the presence of sigma^2, we have in fact assumed the noise variance sigma^2=1  (see second paragraph of section 2.1), which is why we omitted it from the expression.
>
> >In the definition 1, you aim to minimize the variance subject to a constraint on bias. But, in the later (Theorem 2, Figure 1, etc.), it seems to me that you don't mention again, but only mention . At least you should mention what is as a function of or vice versa in Theorem 2.
>
> We have included an explicit description of the relationship between alpha and C in the supplement (A.8), see also our response to reviewer ACbc.
>
> >In Figure 1, you only plot for the very special case when X is diagonal. You should also plot for other cases of X.
>
> Thank you for pointing this out. However, the diagonal case is actually no loss of generality here. For consider any matrix X with svd UDV^T. Given some matrix L that satisfies |LX-I|<C, consider L’=VLU. It is easy to see that |L’D-I|<C and Tr(LL^t)=Tr(L’L’^T). This implies that if we know the optimum L(D) we can just rotate it to obtain L(X).
>
> >Typo in Theorem 2.1, the expectation should be over epsilon
>
> Corrected
>
> Right below Figure 2: Figure 3.1 -> Figure 3.
>
> Corrected
>
> Similarly, Figure 3.2.1-> Figure 5
>
> Corrected

---

> > ### Comment · Reviewer_v15j · 2023-11-18
> > **Reply to the authors' rebuttal**
> >
> > Thank you very much for your answering to my questions and comments.

---

### Official Review · Reviewer_ACbc · 2023-10-30

**Soundness:** 2 fair
**Presentation:** 2 fair
**Contribution:** 2 fair
**Rating:** 5
**Confidence:** 3

**Summary:**

The paper considers a variant of linear regression with constraints placed on a "bias operator". Under this framework, the paper discusses an extension of the Gaussian-Markov theorem, showing empirical and theoretical evidence for its main result, Theorem 2. Later discussions in the paper surround "flatness" and "deepness" of various losses considered within the paper.

**Strengths:**

The paper tackles an interesting class of linear regression models and delivers a thorough presentation of various aspects of the problem, from the problem definition, main theorem statement, to several case studies, all of which help to paint the overall picture of the problem. The constrained setup considered in the paper is also interesting and intuitive. Considering that linear models are a core concept of machine learning, the paper is of sufficient interest to ICLR.

**Weaknesses:**

There are several dimensions of weaknesses presented in the paper:

a. Clarity and overall quality of presentation. The paper does not appear to be carefully edited and revised, with multiple typographical errors in the first paragraph of the introduction alone (examples: "somewht" in line 3, lack of period at end of sentence in line 4, reverted quotation marks on line 5, etc.). The graphs, equations, and tables in latter parts of the paper can also benefit from detailed revisions. These issues surrounding clarity and presentation are not constrained to the first paragraph and can be found throughout the paper and also the supplementary material.

b. Discussion of main theorem. While the problem setting itself is interesting, the discussions of the main theorem (Theorem 2) leave an impression that it can be further discussed. For example, what are the values of alpha? Although it is shown in the supplementary that alpha is a consequence of solving Equation 1 using Lagrange multipliers, it is also unclear how large (or small) the value of alpha is and how it impacts the interpretations of the main result.

c. Considering that one of the paper's main claims is a "flat minima" phenomenon, the paper would benefit from stronger theoretical results (apart from simulation-based arguments) surrounding this claim.

d. Possible typo in main theorem. In the main theorem's statement for the nuclear norm, the result relies on $\max(\Sigma,\alpha)$: should $\Sigma$ be replaced with something like the maximum eigenvalue?

**Questions:**

Several questions were listed in the above "weaknesses" section.

**Details Of Ethics Concerns:**

No ethics concerns were found.

---

> ### Author Response · Authors · 2023-11-13
> **Response to Reviewer ACbc**
>
> Thank you for your helpful comments. Below, find our responses to specific points.
>
> >The paper does not appear to be carefully edited and revised, with multiple typographical errors in the first paragraph of the introduction alone (examples: "somewht" in line 3, lack of period at end of sentence in line 4, reverted quotation marks on line 5, etc.). The graphs, equations, and tables in latter parts of the paper can also benefit from detailed revisions. These issues surrounding clarity and presentation are not constrained to the first paragraph and can be found throughout the paper and also the supplementary material.
>
> Thank you for pointing this out. We have corrected the specific typos mentioned here, as well as a large number of other small typos, see the general response to reviewers as well as the individual responses to the other reviewers.
>
> >While the problem setting itself is interesting, the discussions of the main theorem (Theorem 2) leave an impression that it can be further discussed. For example, what are the values of alpha? Although it is shown in the supplementary that alpha is a consequence of solving Equation 1 using Lagrange multipliers, it is also unclear how large (or small) the value of alpha is and how it impacts the interpretations of the main result.
>
> We have added a discussion of the explicit relation between alpha and C in the appendix (section A.8), with a pointer following the main theorem.
>
> >Considering that one of the paper's main claims is a "flat minima" phenomenon, the paper would benefit from stronger theoretical results (apart from simulation-based arguments) surrounding this claim
>
> We appreciate this concern. While the review period is unfortunately likely too short for us to develop substantial new theoretical results, we can at least make an interesting theoretical observation about our results which bears on the "flat minima" phenomenon, and  was not mentioned in the main paper.  Namely, starting from our integral-based expressions for the test error (Prop. 2.1 and 2.4), we can obtain explicit expressions for the curvature of the error curve simply by differentiating under the integral. The resulting analytic expressions are complicated and not particularly illuminating, which is why we did not include them in the paper, and opted instead to explore the properties of the minima through simulations.
>
> >In the main theorem's statement for the nuclear norm, the result relies on \Sigma should be replaced with something like the maximum eigenvalue?
>
> The result is correct as stated, but we see how the notation could potentially be confusing here. We have clarified that the max(...) refers to an elementwise maximum of the singular values. (Note that we have also changed the notation from \Sigma to \sigma here).

---

### Official Review · Reviewer_cHcu · 2023-10-30

**Soundness:** 3 good
**Presentation:** 3 good
**Contribution:** 3 good
**Rating:** 6
**Confidence:** 4

**Summary:**

The Gauss-Markov theorem says that the best unbiased linear estimator is the pseudoinverse of the data matrix (in the paper denoted by $X$), which can be obtained by minimizing the Frobenius norm of the estimator $L$ subject to the constraint that $L$ is the left inverse of the data matrix, $LX=I$. The paper generalizes this formulation by relaxing this constraint to $||LX-I||_p\le C$ where $||\cdot||_p$ is the Schatten $p$-norm (vector $p$-norm on the singular values), thus allowing some bias $C$. A special case is the ridge/Tichonov regression, obtained for $p=2$. The parameter $C$ (or a monotonically related parameter $\alpha$) is determined by validation, by minimizing the test error. The advantage is that for $p\neq2$, the minimum of the test error over $C$ may be flatter than for $p=2$.

The authors derive a closed form solution for the optimal estimator $L$ for $p=1$ (nuclear norm) and $p=\infty$ (spectral norm). Next, they derive explicit form (as integrals) of for the test error in thermodynamical limit for two special distributions of the data: spherical Gaussians (elements of $X$ and noise in right-hand sides are i.i.d. normal) and diagonal data (when the Gramian $X^TX$ is diagonal). The integrals are solved in closed form for $p=1$ and $p=\infty$. This theoretical formula is shown to agree with test error on synthetically generated data.

The cases $p=1,2,\infty$ are compared in a simulated experiment in which $\alpha$ with smallest test error is estimated by x-validation, where best $\alpha$ is selected by "grid search" from 9 log-spaced values. This showed that the nuclear-norm regression is comparable to or better than (depending on methodology) ridge regression. A similar result is obtained for the similar experiments for nonlinear regressors (random Fourier features).

**Strengths:**

The observation that a wider minima of $\alpha$ can be achieved at the cost of a little bias is interesting.
The theoretical results (test errors) are non-trivial to derive.
The text is clear enough, though clarity could be improved by more effort.

**Weaknesses:**

I cannot assess novelty reliably because my expertise is mainly in optimisation rather than estimation (however, I understand all parts of the main paper well). In fact, rather than extending Gauss-Markov theorem, the paper generalizes ridge regression (please, consider changing the title accordingly). It is well-known that ridge regression has non-zero bias but a smaller variance than pseudoinverse.
So the main novelty seems to be that of flatter minima of test error, rather than the generalization of ridge regression.

Though the minima for the nuclear-norm regression are indeed flatter than for the ridge regression, the difference is sometimes only minor, as seen in Figure 2. It is true that the experiments show that in estimating $\alpha$ by x-correlation, the nuclear norm most often wins. However, this might be due to the experimental methodology. E.g., if there were more than 9 values of $\alpha$, the deeper minima of the ridge regression might have been hit much more often.

A major weakness, in my opinion, is that the experiments are done only on synthetic data. The applications of linear regression are abundant, so it should be possible to find many suitable real datasets for this.

Minor/fixable issues:

1st formula in section 2.1: symbol $L(X)$ is used here but then never more. Change to $L$.

Thm 2: Letter $\Sigma$ is usually used to denote the diagonal matrix with singular values. For vector of singular values, better use $\sigma$ or $s$.

There are many small mistakes/typos in the text. E.g., references to figures in sections 2.2.4 and 3 refer to non-existent figures (e.g., figure 2.2.4 in section 2.2.4).

In the 5th line of section 2.2.1, can the formula for MSE be simplified (i.e., calculate the mean value in closed form)? It is confusing that the MSE has quite different form in sections 2.2.1 and 2.2.2 (this lets the reader wonder if this difference is substantial or just due to little care for text clarity). This might deserve a comment.

Most displayed equations are unnumbered, which is not friendly for reviewers (given that lines are not numbered in the ICLR style).

The asterisk symbol in Proposition 2.1 and in the first displayed formula in section 2.2.2 has not been defined.

The integrals for $Err(\alpha)$ in sections 2.2.1 and 2.2.2 are almost the same, up to the integrating measures. I wonder if they are correct or there are typos in them..?

POST REBUTTAL: The authors have clarified my objections, to certain extent. I am therefore raising my evaluation. However, please note that I cannot assess novelty reliably.

**Questions:**

It would be helpful to vary the number of values of $\alpha$ in the grid method (currently this value is 9) in the experiments, i.e., to make it a hyperparameter. Pls see my remark on this in "weaknesses".

It would be helpful in Figure 5 to report not only winners but also MSE for different models (as in Figure 6 left) - because a winner can win by only a small margin.

Why are any experiments on real data not included? Is there a theoretical obstacle? Or, perhaps, you believe that the results of such experiments would not be informative enough? Please comment (also in the paper, if accepted).

---

> ### Author Response · Authors · 2023-11-13
> **Response to Reviewer cHcu (pt 1)**
>
> Thank you for the careful reading of our paper and helpful comments. Below, find our responses to specific points.
>
> >In fact, rather than extending Gauss-Markov theorem, the paper generalizes ridge regression (please, consider changing the title accordingly). It is well-known that ridge regression has non-zero bias but a smaller variance than pseudoinverse. So the main novelty seems to be that of flatter minima of test error, rather than the generalization of ridge regression.
>
> We appreciate and understand the suggestion to change the title. However, we believe that the characterization of the result in terms of generalizing the GM theorem is accurate and that changing the title would come at the expense of clarity. As we explain in Section 2: “The classic Gauss-Markov theorem tells us that if we want to impose exactly zero bias, then the minimal variance estimator is the ordinary least squares estimator. But what if we want to relax this, to allow non-zero but bounded amount of bias?”. From a presentational standpoint, moreover, we believe that the emphasis of the GM theorem viewpoint is clearer than a framing in terms of generalizing Ridge regression, for the reason that there are many potential ways one could think to generalize Ridge regression, while the GM theorem provides a very specific and principled avenue of generalization.
>
> As per your observation that ridge regression has non-zero bias but smaller variance than OLS, this is certainly true and widely known. But our Theorem 2 in fact implies a significantly stronger (and, we think, more interesting) statement, namely that ridge regression attains the best possible tradeoff between bias and variance, assuming that “bias” is being measured using the Frobenius norm of LX-I (cf. middle panel of figure 1). This particular fact, while easy to prove on its own, does not appear to be well known (at least, we could not find a result to exactly this effect in the literature).
>
> >Though the minima for the nuclear-norm regression are indeed flatter than for the ridge regression, the difference is sometimes only minor, as seen in Figure 2. It is true that the experiments show that in estimating by x-correlation, the nuclear norm most often wins. However, this might be due to the experimental methodology. E.g., if there were more than 9 values of, the deeper minima of the ridge regression might have been hit much more often.
>
> You are certainly correct in observing that the number of alpha values will influence the relative performance of the models in cross-validation. We acknowledged this point, both in section 2.2.3, and in the Discussion: “It should be noted that the particulars of the cross-validation results depend on the number of α values used in the grid search, among other factors; and as noted before, the effect of the curvature at the minimum decreases as more values of α are used.”  That said, we have also included additional results in the Appendix (section A.9) showing the effect of adding more values of alpha. Not surprisingly, the Ridge model often benefits from extra values of alpha in these experiments.
>
> >A major weakness, in my opinion, is that the experiments are done only on synthetic data. The applications of linear regression are abundant, so it should be possible to find many suitable real datasets for this.
>
> We appreciate this concern. When writing the paper, we considered including such experiments, but opted not to for the following reason. Our rationale in designing the experiments was to (1) understand the factors that can influence the performance of the three considered models, and (2) to illustrate the practical consequences of the “flat minima” phenomenon vis a vis cross-validated accuracy. We thus opted for exclusively synthetic data because it is much easier to control the underlying properties of such data compared with real datasets (as per (1)), and such data are already sufficient to illustrate (2).
>
> That said, there is no issue in principle or in practice with applying either the Spectral or Nuclear regressions to real datasets. In order to address your concern, we have done exactly this, and included the results in the Appendix (section A.11). To summarize, we consider the well-known Diabetes dataset and California housing dataset, both available from sklearn. For each dataset, we generated random splits with a training set size of 300, and the remaining samples used for testing. Otherwise, we followed the same methodology as in the synthetic Gaussian simulations. We find results that are qualitatively similar to the table5 from the Gaussian data. Namely, the Nuclear has a small advantage in terms of average MSE, and a fairly substantial one in terms of the probability of winning on a given split.
>
> (cont...)

---

> ### Author Response · Authors · 2023-11-13
> **Response to Reviewer cHcu (pt. 2)**
>
> >1st formula in section 2.1: symbol L(X) is used here but then never more. Change to L
>
> Corrected
>
> >Thm 2: Letter \Sigma is usually used to denote the diagonal matrix with singular values. For vector of singular values, better use \sigma or s
>
> Changed to \sigma
>
> >There are many small mistakes/typos in the text. E.g., references to figures in sections 2.2.4 and 3 refer to non-existent figures (e.g., figure 2.2.4 in section 2.2.4).
>
> Corrected. We have also corrected a large number of other small typos, see the general response to reviewers as well as the individual responses to the other reviewers.
>
> >In the 5th line of section 2.2.1, can the formula for MSE be simplified (i.e., calculate the mean value in closed form)? It is confusing that the MSE has quite different form in sections 2.2.1 and 2.2.2 (this lets the reader wonder if this difference is substantial or just due to little care for text clarity). This might deserve a comment.
>
> Thank you for pointing this out. We have clarified this point in the text- we note that the expression for MSE can be algebraically simplified, but that we have defined it as we did in order to make clear that it is an average over the test distribution. We have also taken the opportunity to further explain why there is a difference between the MSE expressions between the spherical Gaussian and diagonal cases in section 2.2.2. We note there that the difference arises from the fact that the test observations (i.e. rows of Xtest) are not independent in the diagonal case.
>
> >Most displayed equations are unnumbered, which is not friendly for reviewers (given that lines are not numbered in the ICLR style).
>
> We have inserted numbers in all displayed equations in both the main text and supplement
>
> >The asterisk symbol in Proposition 2.1 and in the first displayed formula in section 2.2.2 has not been defined.
>
> This symbol denotes the multiplication of two numbers- however we have removed it for the sake of clarity
>
> >The integrals for ERR in sections 2.2.1 and 2.2.2 are almost the same, up to the integrating measures. I wonder if they are correct or there are typos in them..?
>
> You are correct that these two integrals are nearly the same. The basic reason for this is that both formulas are derived in a similar fashion of exploiting rotational symmetry of the matrix ensemble to reduce the MSE to a sum over the spectrum of Xtr. Besides the integrating measures, the only difference is that the integral for the Diagonal case has an extra factor of x, which intuitively corresponds to the fact that errors in estimating the coefficient for a direction of high variance contributes more to the overall error than for a direction of low variance. We have clarified this point in the main text.
>
> >It would be helpful to vary the number of values of alpha in the grid method (currently this value is 9) in the experiments, i.e., to make it a hyperparameter. Pls see my remark on this in "weaknesses".
>
> We have done this and included the results in the appendix, see above.
>
> >It would be helpful in Figure 5 to report not only winners but also MSE for different models (as in Figure 6 left) - because a winner can win by only a small margin.
>
> In the new results with varied number of alpha (A.9), we also report the individual MSE and win probability for each model. In these results, we considered only a relatively small number of the cells shown in Figure 5, however, we think they give a representative idea of the size of the margin of victory. Indeed, we see that the margin of victory in terms of MSE can be slim, but the win probabilities are usually more decisive (i.e. significantly >33%).
>
> >Why are any experiments on real data not included? Is there a theoretical obstacle? Or, perhaps, you believe that the results of such experiments would not be informative enough? Please comment (also in the paper, if accepted).
>
> See above.

---

> ### Comment · Reviewer_cHcu · 2023-11-15
> **Response to Rebuttal**
>
> Thank you for your response to my review. Let me react to some of them.
>
> Ad the title: This is a minor thing and I agree with the original title. Let me just remark that when initially reading the title, I expected something more. The crucial thing about GM is that the variance is the least, wrt semidefinite the partial order on covariance matrices. We do not have this here (as you mention in the paper). We sacrifice this, together with zero, just like in ridge regression.
>
> Ad larger numbers of $\alpha$: Thanks for appendix A.9. It shows that (expectedly) larger numbers of $\alpha$ makes ridge much better -- but *not always*. Namely, in top-right subfigure in Figure 7 in A.9,  ridge does not win even for 50 values of $\alpha$. This means that either the ridge minimum is really really narrow (so that we would need more values of $\alpha$ to find it -- please consider such experiment!) or that the global minimum of test error is smaller for spectral than for ridge. Which of these two options is true?
>
> Thanks for including the experiment on real data. They confirm the theoretical results.
>
> Let me have one more comment: another reviewer asked about the relation between $\alpha$ and $C$. I wonder, if problem (3) (in the revised manuscript) were reformulated to a penalty form, i.e., we minimize $\\|LX-I\\|_p + {\alpha\over 2}\\|L\\|_F^2$ unconstrained, would this be the same $\alpha$ as in the paper? If so, please consider to include this for better clarity (rather than just section A.8).

---

> ### Author Response · Authors · 2023-11-16
> **Response to response to rebuttal**
>
> Thank you for responding to our rebuttal. To address your new points:
>
> >Namely, in top-right subfigure in Figure 7 in A.9, ridge does not win even for 50 values of alpha . This means that either the ridge minimum is really really narrow (so that we would need more values of to find it -- please consider such experiment!) or that the global minimum of test error is smaller for spectral than for ridge. Which of these two options is true?
>
> The short answer is that the ridge minimum is smaller than for the spectral, but is comparatively very narrow. The panel you point out has rho=0 (i.e. uncorrelated features), so we can use some of the analytic results from the paper to understand this phenomenon. As we remarked in section 2.2.3, the global minimum of Ridge will always be smaller than for the Nuclear or Spectral in this case. Looking at the bottom rows of Figure 3, on the other hand, we see that the Spectral (but not Nuclear) has much flatter minima than Ridge for large sigma and intermediate lambda (for reference, the panel in question has sigma=3.5 and lambda=.5). This is entirely consistent with what we see in the panel from A.9-where the Spectral, but not Nuclear, outperforms Ridge in this case even for large numbers of alpha.
>
> It is also important to note that, even though the ground truth error curve for Ridge may have the lowest global minimum, we never actually directly observe this curve (since it is an average over all datasets, and taken in the thermodynamic limit). Rather, we only estimate the value for any fixed alpha by using cross-validation. The selection of random CV splits introduces noise into our MSE estimates. If the scale of this noise is comparable to the difference between the Ridge minimum and Spectral minimum, then adding more alpha values can result in "overfitting to the validation set", where we select an alpha value that happened to get a lucky CV split over one that is actually close to the minimum.  So it is possible that even with large numbers of alpha we will not find the global minimum with this methodology.
>
>
> >Let me have one more comment: another reviewer asked about the relation between and . I wonder, if problem (3) (in the revised manuscript) were reformulated to a penalty form, i.e., we minimize ${\frac {\alpha}2}\||L\||_2^2+\||LX-I\||_p$ unconstrained, would this be the same as in the paper? If so, please consider to include this for better clarity (rather than just section A.8).
>
> Unfortunately, the alpha in this constrained problem would not exactly correspond to the alpha in Theorem 2. We can illustrate this most easily in the ridge case. There, we can use matrix calculus to see that the solution L to the penalty problem must satisfy $\alpha*L+{\frac {(LX-I)X^T}{ 2||LX-I||}}=0$.After some algebra, we see that this corresponds to Ridge regression with a regularization strength of $2||LX-I||\alpha$

---

### Author Response · Authors · 2023-11-13
**General response to reviewers**

Thank you to all of the reviewers for the constructive and helpful feedback. We have uploaded a new version of the manuscript and supplement in response to the comments. Moreover, since all three reviewers commented on the number of small typos and mistakes in the original manuscript, we have gone through and tried to find and correct as many as possible. Below, we record the typos we have corrected in the newly uploaded manuscript, not inclusive of the specific ones pointed out in the reviews (which we have also corrected). We welcome the reviewers to point out any further ones that we may have missed.

Pg 1: Rank-> rank

Pg 1: Norm-> norm

Pg 1: Linear-> linear

Pg 1: settting-> setting

Pg 2: significanly-> significantly

Pg 2: MSE-> test error

Pg 2: Regression-> regression

Pg 2: add condition p>=1 to definition of Schatten norm

Pg 4: exogeneous->exogenous

Pg 4: MP-> Marchenko Pastur

Pg 4: cdf-> cumulative distribution function

Pg 4: note-> not

Pg 6: is Hessian at the minimum -> is the Hessian  of $Err(\alpha)$ at the minimum

Pg 6: Figre->Figure

Pg 6: corresopnding-> corresponding

Pg 6: corespondence-> correspondence

Pg 8: Fix backwards quote  (“two different definition of “best””)

Pg 8: aggreate-> aggregate

Pg 8 add definition of RFF acronym

Pg 8 add period to footnote 6

Pg 8: rff-> RFF

Pg 8 fourier -> Fourier

Pg 9: colinearity-> collinearity

Pg 9: Componnts-> Components

Pg 9: add definition of PCR acronym

Pg 9: regularizaitons-> regularizations

Pg 10: testcase-> test case

Pg 10: transfering-> transferring

A.3 :Switched to boldface font for probability and expectation operators throughout

A3:mins-> minima

A3: Add period to very last sentence

Pg 16: psd-> nonnegative definite

Pg 18: add period (“For this, we introduce a Lagrange multiplier…”)

Pg 18: appropraite-> appropriate

Pg 18: ((-> (

Pg 19: indepdent-> independent

Pg 19: add missing period (“where the cross term vanishes..”)

Pg 19: use boldface for expectation operator (same location)

A.7 title: mins-> minima

Pg. 20: E-> Err

Corrected all issues with Figure referencing

---

### Author Response · Authors · 2023-11-22
**Kindly reminder- discussion period closing**

Dear reviewers,

First, we would like to thank you again for your valuable feedback and time spent reviewing our paper. We believe the paper has been significantly strengthened as a result of incorporating your feedback.

With the discussion period closing today, we are looking forward to your continued engagement. We are particularly keen to hear whether you feel our revisions have sufficiently addressed your concerns, and if not then why not.

Please also let us know if you have any further comments or questions regarding the paper.

Regards,

Authors

---

### Meta-Review · Area_Chair_eEtc · 2023-12-07

**Metareview:**

This paper discusses an extension of the Gauss-Markov theorem for linear estimation, allowing for a non-zero but bounded bias operator with a specific matrix norm constraint. The paper presents the derivation of optimal estimators for nuclear and spectral norms, including the recovery of ridge regression for the Frobenius case. It also mentions the analysis of generalization error in random matrix ensembles and a simulation study demonstrating that nuclear and spectral regressors can outperform Ridge regression in various scenarios.

The paper received three reviews, and it was considered borderline. One reviewer pointed out a mismatch between the paper's title and its actual contributions, suggesting a title change. Another reviewer mentioned a lack of discussion on the main theorems, which the authors addressed in the revised version. Despite these valid comments, the paper's contributions are seen as valuable, offering a fresh perspective on extending the Gauss-Markov Theorem. While not a strict extension, it presents new and interesting results. Therefore, I recommend acceptance, with the hope that the authors will address the reviewers' feedback in their revisions.

**Justification For Why Not Higher Score:**

The reviewers had several reasonable concerns.

**Justification For Why Not Lower Score:**

Although the reviewers raised several concerning comments, the overall consensus is that the novelty of the paper outweighs its shortcomings.

---

### Decision · Program_Chairs · 2024-01-16

Accept (poster)